# Comparison of Hybrid Posterior Fixation and Conventional Open Posterior Fixation Combined with Multilevel Lateral Lumbar Interbody Fusion for Adult Spinal Deformity

**DOI:** 10.3390/jcm11041020

**Published:** 2022-02-16

**Authors:** Hirooki Endo, Hideki Murakami, Daisuke Yamabe, Yusuke Chiba, Ryosuke Oikawa, Hirotaka Yan, Minoru Doita

**Affiliations:** Department of Orthopedic Surgery, Iwate Medical University, 2-1-1 Idaidori, Yahaba-cho, Shiwa 028-3695, Japan; oki_oki@me.com (H.E.); mehelen7618@gmail.com (D.Y.); yusuke.c.0718@gmail.com (Y.C.); ryosuke_oikawa@yahoo.co.jp (R.O.); m.yanhir@gmail.com (H.Y.); doita@iwate-med.ac.jp (M.D.)

**Keywords:** adult spinal deformity, minimally invasive surgery, lateral lumbar interbody fusion, anterior and posterior combined surgery, percutaneous pedicle screw, surgical outcome

## Abstract

We compared radiological and clinical outcomes between multilevel lateral lumbar interbody fusion (LLIF) + hybrid posterior fixation (PF) and multilevel LLIF + conventional open PF in patients with adult spinal deformity (ASD). Patients who underwent minimally invasive surgery for ASD in a single institution between 2014 and 2018 were retrospectively reviewed. Fifty-six patients (hybrid PF, 30; open PF, 26) who underwent ASD correction surgery were enrolled between 2014 and 2018. We evaluated patients’ demographics, clinical outcomes, and radiographical parameters in each group. There was significantly less estimated blood loss in the hybrid PF group (662.8 mL vs. 1088.8 mL; *p* = 0.012). The CRP level 7 days after surgery was significantly lower in the hybrid PF group (2.9 mg/dL vs. 4.3 mg/dL; *p* = 0.035). There was no significant difference between the two groups in other demographic variables, visual analog scores for back pain and leg pain, Oswestry Disability Index, coronal Cobb angle, lumbar lordosis, pelvic tilt, pelvic incidence–lumbar lordosis mismatch, and sagittal vertical axis. There was a significantly higher percentage of major complications in the open PF group (42.3% vs. 13.3%; *p* = 0.039). Thus, LLIF + hybrid PF for ASD corrective surgery may be comparable to LLIF + open PF in terms of clinical and radiographic outcomes.

## 1. Introduction

Adult spinal deformity (ASD) is defined as the presence of at least one of the following: (1) sagittal vertical axis (SVA) of ≥5 cm, (2) pelvic tilt (PT) of ≥25°, (3) pelvic incidence (PI)–lumbar lordosis (LL) mismatch of ≥10°, coronal Cobb angle of ≥20°, and thoracic kyphosis of ≥60° [1].

ASD often causes gait disturbance and difficulty in standing due to severe lower back pain, leg pain, neural deficit, gastrointestinal disorders such as gastroesophageal reflux disease, and respiratory dysfunction. Moreover, decreased lumbar lordosis and worsening of the sagittal alignment of the spine, including the pelvis, reduce health-related quality of life (HRQOL) [2].

ASD has an estimated prevalence of 6% in people over 50 years of age, and the prevalence of ASD in elderly patients was shown to exceed 60% [3]. It is expected that the prevalence of ASD will increase as the elderly population increases, and the importance of treating this disease will become greater.

The main pathology of ASD is degeneration and deformity at the disc level. Therefore, posterior lumbar interbody fusion or transforaminal lumbar interbody fusion (TLIF) is performed as the conventional mainstream corrective surgery, and three-column osteotomies such as pedicle subtraction osteotomy (PSO) and vertebral column resection are performed for severe and rigid deformities, including osteoporotic vertebral body fractures. While these conventional procedures have achieved stable postoperative outcomes with the development of surgical techniques and instrumentation, they are highly invasive and could lead to serious complications such as intraoperative massive bleeding, neurological deficit, postoperative wound infection, and pulmonary embolism.

Complications are reported to occur in more than 40% of ASD patients who undergo open surgery. Risk factors for increased complication rates include patient age and medical comorbidities [4]. Therefore, surgical indications for elderly ASD patients with medical comorbidities are limited.

In recent years, minimally invasive surgery (MIS) represented by lateral lumbar interbody fusion (LLIF) emerged as a surgical technique to reduce the risks associated with conventional procedures, and it was gradually applied to ASD. There are multiple reports about its effectiveness [5,6]. Lai et al. reported that overall and intervertebral stability significantly increased after multilevel LLIF and that bilateral pedicle screws provided the greatest stability, followed by a unilateral pedicle screw and lateral plate, in the cadaveric biomechanical analysis [7]. In addition, minimally invasive posterior surgery with the use of percutaneous pedicle screws (PPSs) was used to prevent damage to the paraspinal muscles, which was a serious problem with the conventional approach [8].

Several techniques involving open posterior procedures combined with minimally invasive anterior techniques using LLIF (hybrid surgeries) are widely performed. In recent years, further minimally invasive posterior techniques using PPSs combined with LLIF (cMIS) were developed depending on the severity of the case, and MIS has rapidly gained widespread popularity as a surgical procedure for spinal deformity.

Focusing on minimally invasive techniques, deformity correction potential, and the indirect decompression effect of LLIF, we performed corrective surgeries incorporating LLIF for patients with ASD.

Over time, surgical procedures and instrumentations were developed, and equipment such as PPSs and rods that can be used concomitantly with conventional open techniques were introduced, allowing for minimally invasive corrective fixation for spinal deformities in the elderly, which was previously difficult with cMIS. We developed a new surgical technique, multilevel LLIF + hybrid posterior fixation (PF), which combined the conventional open procedure with MIS using PPSs for severe spinal deformity such as kyphoscoliosis in the elderly, with satisfactory results (Figure 1).

Patients with symptomatic ASD underwent multilevel LLIF + conventional open PF between 2014 and 2015 and multilevel LLIF + hybrid PF between 2016 and 2018 in our institution. A similar procedure to hybrid PF was subsequently reported by Katz et al. in 2019 and was found to have good clinical and radiographic results [9]. However, to our knowledge, there are no reports comparing postoperative results between hybrid PF and open PF. Therefore, the objective of this study was to evaluate and compare the radiological and clinical outcomes of multilevel LLIF + hybrid PF with those of multilevel LLIF + conventional open PF for ASD.

## 2. Materials and Methods

A database of ASD patients who underwent minimally invasive surgery for ASD at a single institution between 2014 and 2018 was retrospectively reviewed. The inclusion criteria were as follows: (1) age > 50 years at the time of surgery, (2) history of anterior corrective fusion using multilevel LLIF over 3, (3) upper instrumented vertebra cranial to ≥T10, (4) distal sacropelvic fixation using S2 alar iliac screw (S2AIS), and (5) minimum of 2 years of follow-up. Patients with previous spinal instrumentation, acute spinal trauma, spinal tumor, spinal infection, severe spinal deformities requiring Schwab grade 3–6 osteotomy [10], and circumferential bony fusion across multiple vertebrae were excluded. A total of 56 patients who underwent ASD correction surgery met the inclusion criteria (mean age at surgery 67.9 years; 14 men and 42 women). All patients underwent ASD correction surgery by the same three experienced spine surgeons who belonged to the institution.

The included patients were classified into two groups: 30 in the hybrid PF group and 26 in the open PF group. Demographic and operative parameters were collected, including age at the time of surgery, sex, bone mineral density (BMD) of the femoral neck (young adult mean), number of interbody fusion levels, number of posterior fusion levels, operative time, and intraoperative and postoperative blood loss through a drainage tube until 2 days post-surgery. C-reactive protein (CRP) levels were also recorded before and 1, 3, and 7 days after surgery.

Clinical outcomes were assessed using the visual analog scale (VAS) for back pain and leg pain and the Oswestry Disability Index (ODI) as HRQOL questionnaires. ODI scores were based on a scale of 0–100. A score of 0–20 indicated minimal disability; 20–40, moderate disability; 40–60, severe disability; 60–80, crippled; and 80–100, bedbound or exaggeration of symptoms [11]. These questionnaires were administered before and 2 years after surgery.

Radiographic outcomes were assessed using standing radiographs for all patients before surgery, after surgery, and at the final follow-up. Coronal and sagittal parameters were defined as subsequently outlined. The degree of scoliosis was measured by the coronal Cobb angle; and lumbar and thoracic curves were measured between the most coronally angulated vertebral bodies. LL was the angle between the L1 and S1 superior endplates. PI was defined as the angle between a line perpendicular to the S1 superior endplate and a line connecting the midpoint of the S1 superior endplate to the center of the femoral head. PI-LL mismatch was defined as PI minus LL. PT was the angle between a vertical line and a line connecting the midpoint of the S1 superior endpoint to the center of both femoral heads. SVA was defined as the distance from the posterosuperior aspect of S1 to the C7 plumb line.

Major or minor complications were categorized according to the classification of Glassman et al. [12], who investigated perioperative complications after ASD surgery. All patients who underwent ASD surgery in this study were investigated during and after surgery based on this classification.

### 2.1. Surgical Procedure

All patients (both hybrid and open PF groups) initially underwent LLIF surgery. LLIF was performed in the lateral recumbent position. To achieve satisfactory curvature of the lumbar spine, LLIF was performed on as many vertebral bodies as possible, extending above the highest wedge-shaped vertebral body that could cause spinal stenosis and intervertebral disk degeneration. Generally, LLIF was performed on three to four vertebral bodies, extending from L2/3 (or L1/2) to L4/5. Subsequently, posterior fusion surgery was performed in both groups according to their respective procedures.

#### 2.1.1. Hybrid PF

Using a posterior approach with the patient in the prone position, PPSs were inserted in the upper aspect of LLIF, usually from L2 to S1. An S2AIS was also inserted percutaneously using the PPS through a 3 cm midline skin incision at the level of the S1 spinous process. LLIF could not be performed at L5–S1 in general; thus, mini-open TLIF was usually performed on both sides using the posterior approach. At the superior aspect of LLIF, the thoracic region required bone grafting; as such, the posterior approach for an open procedure was conventionally performed to insert the open pedicle screw. If necessary, Ponte facetectomy was concomitantly performed to allow better correction of the alignment. Rods bent a little more than the target degree of the lumbar curvature were inserted beneath the muscle layer from the cranial side (the most caudal end of the skin incision in the thoracic approach from the open approach). After going through the blade of the PPS, the set screws were placed sequentially from the S2AIS toward the cranial side. As the optimally bent rods were gently inserted, the corrective position was set with the rod rotation maneuver and cantilever technique (Figure 2). After the surgery, the patients were allowed to walk with a hard corset, which they were required to wear for 3 months after surgery, followed by a soft corset for another 3 months, while avoiding motions such as bending and twisting in their daily activities.

#### 2.1.2. Open PF

Using a posterior approach with the patient in the prone position, the pedicle screws and S2AIS were inserted using the open technique from the thorax to the pelvis. Ponte facetectomy was concomitantly performed to allow better correction of the alignment as needed. Subsequently, TLIF was usually performed on the left or right side to reduce invasion at the L5–S1 level. As the optimally bent rods were gently inserted, the corrective position was set with the rod rotation maneuver and cantilever technique. After the surgery, the patients were allowed to walk with a hard corset, which they were required to wear for 3 months after surgery.

### 2.2. Statistical Analysis

Differences between preoperative and postoperative parameters were analyzed using a paired t-test. The chi-square and Mann–Whitney U tests were used to assess significant differences between the two groups, and *p* < 0.05, with a confidence interval of 95%, was considered statistically significant. All statistical analyses were performed using SPSS software version 26.0 (IBM, Armonk, NY, USA).

### 2.3. Results

The demographic and operative parameters of the two groups are summarized in Table 1. There was no significant difference in the age at surgery between the two groups, with a mean of 68.2 years for the hybrid PF group and 67.6 years for the open PF group. There was also no significant difference in sex and BMD at the femoral neck. However, the open PF group had a significantly longer follow-up period than the hybrid PF group (*p* = 0.004).

There was no significant difference in the operative time between the two groups. However, intraoperative and postoperative blood losses were significantly lesser in the hybrid PF group than in the open PF group (*p* = 0.012 and *p* = 0.001, respectively). The CRP level 1 day after surgery was significantly higher in the hybrid PF group than in the open PF group (*p* = 0.019). The maximum postoperative serum CRP levels were observed on day 3 after surgery, with no significant difference. On day 7 after surgery, the CRP level was significantly lower in the hybrid PF group (*p* = 0.035). The mean number of levels was 4.3 for interbody fusion and 10.3 for posterior fusion in the hybrid PF group, and 4.0 for interbody fusion and 10.5 for posterior fusion in the open PF group (Table 1).

There was a significant improvement in all radiographic parameters (*p* < 0.001), including the coronal Cobb angle, LL, PI-LL mismatch, PT, and SVA (Table 2) in the hybrid PF group. With regard to clinical outcomes in the hybrid PF group, the mean VAS scores for back pain and leg pain significantly improved (*p* = 0.036 and *p* = 0.011, respectively) 3 months after surgery, and the improved scores were maintained at the final follow-up. The mean ODI also significantly improved at 3 months after surgery (*p* = 0.016), with further improvement at the final follow-up (Table 3). The open PF group also showed significant improvement in all radiographic parameters (*p* < 0.001; Table 2). With regard to clinical outcomes, the mean VAS scores for back pain and leg pain significantly improved (*p* < 0.001 for both) 3 months after surgery, and they were maintained at the final follow-up (Table 3). The mean ODI also significantly improved at 3 months (*p* = 0.001), with further improvement at the final follow-up (Table 3).

On comparison of radiographic and clinical parameters between the two groups, the hybrid PF group showed a larger SVA than the open PF group before surgery, although the difference was not significant. In addition, there were no significant differences between the two groups in terms of all preoperative and postoperative radiographic parameters, including SVA. Similarly, there were no significant differences between the two groups in terms of preoperative and postoperative VAS scores for back pain and leg pain and ODI.

The rate of major complications was significantly higher in the open PF group (42.3%) than in the hybrid PF group (13.3%; *p* = 0.039). In the hybrid PF group, two patients had a rod fracture: 16 months after surgery at the lumbosacral junction in one and 51 months after surgery at the LLIF segment in the other. One patient had a neurological deficit due to the backout of the TLIF cage at the L5–S1 level. Segmental vessel injury occurred in one patient. Reoperation was performed for all patients who had major complications, and all of them recovered.

In the open PF group, seven patients had rod fracture at the following locations: the lumbosacral junction in five (mean 31 (7–78) months); 14 months after surgery at the level of the anterior longitudinal ligament, with rupture during LLIF surgery in one; and 51 months after surgery at the LLIF segment in one. Two patients developed deep wound infection perioperatively, while one patient experienced late wound infection 30 months after surgery. Reoperation was performed in six patients with major complications, and all of them recovered. One patient could not undergo reoperation because of a poor general condition.

## 3. Discussion

For patients with ASD, it is widely known that improvements in the sagittal spinopelvic alignment and whole spine alignment, which are radiographic parameters, are correlated with improved clinical outcomes [13], and corrective surgeries can be conducted to primarily target the following: PT < 20°, PI-LL mismatch < 20°, and SVA < 50 mm as the optimal spinopelvic alignment parameters [14].

In recent years, surgery for ASD has rapidly developed with the development of MIS. Compared with conventional procedures, cMIS, which is a combination of LLIF and PPS, has led to reductions in blood loss and complications and has allowed for early patient recovery [15]. However, sagittal plane correction is inadequate with cMIS, and the procedure is unsuitable for patients with severe sagittal plane misalignment. Anand et al. reported a ceiling effect in cMIS, whereby achievement of a PI-LL of <10° required a preoperative PI-LL mismatch of <38°, and achievement of a SVA of <50 mm required a preoperative SVA of <100 mm [16].

Mummaneni et al. proposed the minimally invasive evaluation and treatment for adult degenerative deformity (MiSLAT) algorithm [17], which included MIS in the selection of surgical procedures for ASD. Subsequently, they proposed a further simplified minimally invasive spinal deformity surgery (MISDEF) algorithm based on the Scoliosis Research Society–Schwab classification modifier [18]. They reported that there was a limitation in the correction of adult spinal deformity by MIS and that cases with SVA ≥ 7 cm, PT ≥ 25°, LL-PI mismatch ≥ 30°, and thoracic kyphosis ≥ 60° on a preoperative plain radiograph should be managed by a conventional procedure.

With the progress of minimally invasive techniques, Mummaneni et al. proposed the MISDEF2 algorithm, which was further updated by the addition of various techniques such as a hybrid-open approach and cMIS using anterior column resection, mini-open PSO, and surgery using an expandable cage [19]. The authors stated that MIS surgery is not recommended and that open deformity surgery should only be performed when patients have undergone prior surgery with instrumentation that required revision or when patients have undergone instrumentation of five levels or more during prior fusion, including L5–S1. Such patients have significant sagittal plane abnormalities and require instrumentation of more than 10 segments. Currently, the surgical procedure for ASD is selected according to the MISDEF2 algorithm in our institution, and LLIF + hybrid PF is classified as class III in the MISDEF2 algorithm.

A few reports have compared radiographic and clinical outcomes between LLIF + open PF and cMIS. Haque et al. reported that LLIF + open PF resulted in significantly greater LL correction than cMIS (*p* = 0.045) and that the mean change in PI-LL mismatch was larger with LL + open PF (*p* =0.003) [20]. Regarding clinical outcomes, Chan et al. reported that the VAS score for leg pain was lower after cMIS than after LLIF + open PF (*p* = 0.032) [21]; however, no significant difference in clinical outcomes was found in other studies [20,22].

To our knowledge, no report has compared radiographic and clinical outcomes between LLIF + hybrid PF and LLIF + open PF. In the present study, multilevel LLIF with hybrid PF and open PF achieved a similar level of improvement in spinopelvic alignment, not only in terms of the coronal Cobb angle but also in terms of sagittal parameters, including LL, PI-LL mismatch, PT, and SVA. The techniques mostly achieved the target correction proposed by Schwab et al. In addition, hybrid PF and open PF showed the same degree of improvement in clinical parameters. We believe that our minimally invasive procedure can produce satisfactory corrective fixation for ASD that is more severe than that proposed in the MISDEF algorithm.

However, it was reported that spinopelvic parameters differ between races, with Asians having a greater compensatory mechanism in the thoracic spine and pelvis compared with that in the Western population [23,24]. Spinal alignment changes with age, and PT, PI-LL mismatch, and SVA all increase. It was also reported that spinal alignment satisfying Schwab’s formula only applies to the age of 45–64 years [25], and an appropriate LL setting according to PI is required for spinal deformity correction in elderly people aged ≥65 years. There are also some reports on the ideal LL based on age [26,27], and it is anticipated that a minimally invasive treatment algorithm for ASD that takes into account different races and age groups will be developed.

Taneichi et al. compared a conventional method (three-column osteotomy via the posterior open technique) and a minimally invasive hybrid method (LLIF and the posterior open technique) for ASD and reported that the operative time was not significantly different (7.6 vs. 7.8 h), whereas the estimated blood loss (EBL) with the hybrid method (1048 mL) was approximately half that with the conventional method (2095 mL) [28]. Furthermore, when LLIF + open PF and cMIS were compared, the operative times was 700 min and 450 min, with EBL of 1500–2000 mL and 500 mL, respectively [20,29]. Regarding EBL, the hybrid PF group in the present study showed findings comparable with those for cMIS in previous reports. Therefore, hybrid PF may be a beneficial MIS in terms of EBL. Koike et al. reported that the serum CRP levels were significantly higher at 1 and 3 days after LLIF + PPS than after TLIF for degenerative spondylolisthesis; however, the levels were equivalent on day 5 [30]. Control of bleeding from the back muscles during PPS placement is often difficult and can cause a postoperative hematoma, which might cause elevation of the CRP level in the early postoperative period. In this study, similar changes in CRP were observed 1 day after surgery, although the level was significantly lower in the hybrid PF group than in the open PF group 7 days after surgery. This could be attributed to the lesser posterior invasion in the hybrid PF group.

A few reports have compared complications between LLIF + open PF and cMIS and showed that the complication rate was higher in LLIF + open PF than in cMIS [21,22]. Katz et al. reported that the occurrence of complications was associated with the open posterior portion, not with the number of treated LLIF levels [9]. In the present study, the complication rate was significantly higher in the open PF group than in the hybrid PF group. We attribute hybrid PF’s lower complication rate to several overlapping factors. First, hybrid PF’s lower invasiveness compared to open PF contributed to lower rates of infection: in addition, since the procedure was performed by the same surgeons from 2014 to 2018, our results are somewhat biased by a technical learning curve over time. Finally, the hybrid PF technique performed in the second half of this period was upgraded based on several lessons learned from our first-half experiences with open PF, in addition to being minimally invasive, such as changing the TLIF cage insertion at L5/S from unilateral to bilateral and modifying the rod diameter and material.

Rod fracture is a major problem associated with corrective fixation of ASD. Lertudomphonwanit et al. reported that the prevalence of rod fracture after ASD surgery was 18.4%, with a rate of 37% for bilateral rod fractures. Greater preoperative SVA, greater preoperative thoracolumbar kyphosis, increased the number of fused vertebrae in patients who received recombinant human bone morphogenetic protein-2 <12 mg per fused level and the use of a 5.5 mm cobalt chromium (CC) rod was associated with rod fracture. Less improvement in patient satisfaction and self-image were noted in patients with rod fracture [31].

In our facility, rod fracture occurred in 6.7% of patients in the hybrid PF group and in 26.9% of patients in the open PF group; thus, the rod fracture rate was particularly high in the open PF group. The most probable reason for this was the surgical technique, although the open PF group had a long postoperative observation period. Most rod fractures occurred in the lumbosacral junction, and at the time of reoperation, bone fusion at the L5- S1 level was not achieved with TLIF in all cases. Therefore, we developed technical improvements to address this problem when transitioning to the hybrid PF procedure by inserting TLIF cages from both sides for sufficient bone grafting and preserving and stabilizing the posterior bone element as much as possible. Moreover, the rod diameter and material were changed to a 6 mm diameter and titanium alloy, respectively.

Titanium alloy rods are preferred because they are less stiff than CC rods. For a rod to pass through the blades of the PPS inserted in a wide area, especially the blades of the S1 PPS and S2AIS, the rod needs to be bent to a certain extent, and titanium alloy rods are superior in this regard. In addition, if the Young modulus of the rod stiffness differs significantly from that of the bone (bone: 10−30 GPa; titanium alloy: 110 GPa; CC: 200–300 GPa), stress shielding of the bone can cause instrumentation failures such as screw loosening and dislocation [32]. We believe that this helped reduce the incidence of rod fractures in the hybrid PF group. However, the observation period in the hybrid PF group was short, and longer observation is needed in the future.

In addition, the importance of forming two-third lordosis in the lower lumbar spine during ASD correction was reported, and the association between lower lumbar lordosis and clinical results as well as the occurrence of implant-related complications wasnoted [33]. The Global and proportion (GAP) score proposed by the European Spine Study group indicates the risk of postoperative implant-related mechanical complications such as rod fractures. The lordosis distribution index (LDI), which indicates the proportion occupied by the lower arc of LL (L4–S1) in relation to the total LL (L1–S1), is one of the parameters of the GAP score, and an LDI of 50–80% is considered favorable. In the present study, the LDI was 51%, which was at the lower limit despite being a tolerable value for lordosis formation in the lower lumbar spine, and further studies on this are needed in the future. To obtain more favorable lordosis in the lower lumbar spine with MIS, along with better bone fusion and fewer implant-related complications, it is necessary to further develop MIS procedures by measures such as the introduction of oblique lateral interbody fusion at L5–S1 (OLIF51).

However, it should be noted that the basics of spinal deformity surgery are as follows: (1) achievement of good spinopelvic alignment and balance, (2) elimination of local deformity and instability, (3) strong fixation without correction loss, and (4) achievement of reliable bone fusion. These are crucial for obtaining good surgical results in the long term. To obtain these results, it is necessary to apply a strong corrective force to the deformed spine by making full use of various corrective techniques such as compression, distraction, translation, and derotation. Sufficient bone grafting is also required to obtain good bone fusion, but it is not yet possible to perform this in cMIS to the same degree as that in the open procedure. Further development of MIS is also important from the perspective of social circumstances, given the increase in the number of elderly patients and the increase in medical expenses. However, the basics of spinal deformity surgery must be considered, and the unjustified use of minimally invasive techniques should be avoided.

This study had some limitations. First, the sample size was small, and the study was performed at a single institution because multilevel LLIF with hybrid PF is still unique to our institution. Further, the study was retrospective. Second, the patients in the open PF group underwent surgery from 2014 to 2015, whereas those of the hybrid PF group underwent surgery from 2016 to 2018 because we introduced multiple LLIF with hybrid PF in 2016. Third, there was a significant difference in the follow-up period between the two groups. Thus, several important differences, such as those in the complication rate, were detected between the two groups. Appropriately designed studies without these limitations are needed to compare the effectiveness of the two techniques.

## 4. Conclusions

In the present study, multilevel LLIF with hybrid PF was able to achieve radiographic and clinical outcomes comparable to those of multilevel LLIF + open PF. In addition, the amount of intraoperative and postoperative bleeding was significantly reduced, and the incidence of complications also decreased. MIS for ASD is still under development in terms of surgical techniques and instruments, and efforts to develop MIS techniques and improve the surgical outcome need to be made in the future. Our approach requires no special surgical instruments and does not demand high technical skills, which are occasionally required by cMIS. It is considered a useful technique that may be widely used as MIS in many cases of ASD, except those showing severe spinal deformities requiring three-column osteotomy in grades 3–6 of the Schwab osteotomy classification or circumferential bony fusion across multiple vertebrae. 

## Figures and Tables

**Figure 1 jcm-11-01020-f001:**
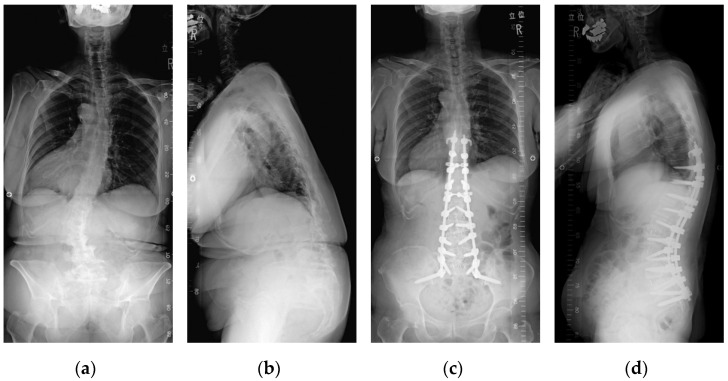
A 70-year-old woman with ASD. Preoperative standing anteroposterior (**a**) and lateral (**b**) radiographs revealed severe spinopelvic imbalance as follows: coronal Cobb angle 46°, LL 7°, PT 51°, PI-LL mismatch 55°, SVA 166 mm, and C7 coronal plumb line 24 mm to the right. The patient underwent T9 to sacropelvic fixation with multilevel LLIF + posterior hybrid PF. Standing anteroposterior (**c**) and lateral (**d**) radiographs at 2 years after surgery show corrected spinopelvic parameters as follows: coronal Cobb angle 9°, LL 56°, PT 20°, PI-LL mismatch 4°, SVA 37 mm, and C7 coronal plumb line 13 mm to the left.

**Figure 2 jcm-11-01020-f002:**
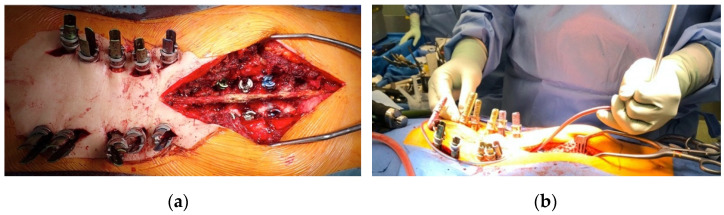
Intraoperative imaging demonstrating hybrid PF for ASD. The PPSs were inserted into the L2 to the ilium, and mini-open TLIF was performed on both sides at L5–S1. Thereafter, open posterior-lateral fusion was performed on T10–L1 using the open pedicle screws (**a**). The optimally bent rods were inserted beneath the muscle layer from the cranial side. After going through the blade of the PPS, the set screws were placed sequentially from the S2AIS toward the cranial side with the rod rotation maneuver and cantilever technique (**b**).

**Table 1 jcm-11-01020-t001:** Demographic and surgical data of patients in the hybrid PF and open PF groups.

		Hybrid PF	Open PF	*p*-Value
No. of patients		30	26	
Age at surgery (years)		68.2 ± 6.9	67.6 ± 5.5	0.744
Sex (Male: Female)		7:23	7:19	0.757
BMD (young adult mean), FN (%)		79.2 ± 13.5	77.4 ± 17.8	0.729
Follow-up after surgery (months)		51.1 ± 15.1	66.7 ± 24.2	0.004
No. of interbody fusion (levels)		4.3 ± 0.7	4.0 ± 0.6	0.175
No. of posterior fusion (levels)		10.3 ± 1.7	10.5 ± 2.1	0.682
Operative time (minutes)		449.2 ± 97.2	464.8 ± 100.7	0.574
Intraoperative blood loss (mL)		662.8 ± 432.3	1088.8 ± 466.3	0.012
Postoperative blood loss (mL)		499.7 ± 226.3	840.3 ± 341.1	0.001
CRP (mg/dL)	Preoperative	0.1 ± 0.2	0.1 ± 0.1	0.726
	Postoperative day 1	9.6 ± 3.4	7.6 ± 2.2	0.019
	Postoperative day 3	14.9 ± 5.5	15.7 ± 6.3	0.632
	Postoperative day 7	2.9 ± 1.7	4.3 ± 2.6	0.035

Values are presented as n or mean ± standard deviation, PF, posterior fixation; BMD, bone mineral density; FN, femoral neck; CRP, C-reactive protein.

**Table 2 jcm-11-01020-t002:** Radiographic parameters of the hybrid PF and open PF groups.

	Hybrid PF	Open PF	*p*-Value
Coronal Cobb angle (°)			
Before surgery	27.8 ± 14.5	32.3 ± 11.9	0.220
After surgery	8.0 ± 6.4	9.5 ± 13.8	0.638
Correction value	18.7 ± 12.7	22.9 ± 11.2	0.207
Final follow-up	7.5 ± 6.1	9.4 ± 5.1	0.532
LL (°)			
Before surgery	11.5 ± 19.7	15.2 ± 22.2	0.538
After surgery	45.7 ± 12.2	46.8 ± 7.7	0.713
Correction value	34.2 ± 19.0	31.5 ± 17.3	0.610
Final follow-up	43.2 ± 11.1	44.5 ± 12.5	0.710
PI-LL mismatch (°)			
Before surgery	41.0 ± 15.7	37.5 ± 20.5	0.507
After surgery	6.2 ± 13.2	5.1 ± 11.9	0.754
Correction value	34.8 ± 19.5	32.7 ± 18.8	0.695
Final follow-up	9.5 ± 13.1	10.0 ± 12.1	0.880
PT (°)			
Before surgery	32.8 ± 9.6	33.8 ± 12.3	0.767
After surgery	22.6 ± 11.2	22.5 ± 9.3	0.972
Correction value	10.2 ± 8.6	10.8 ± 10.5	0.842
Final follow-up	26.1 ± 10.5	27.3 ± 10.3	0.703
SVA (mm)			
Before surgery	138.7 ± 58.2	111.5 ± 56.0	0.099
After surgery	34.3 ± 31.4	32.3 ± 40.2	0.847
Correction value	104.4 ± 52.3	78.2 ± 61.7	0.116
Final follow-up	51.1 ± 43.3	39.5 ± 37.2	0.330

Values are presented as mean ± standard deviation. PF, posterior fixation; LL, lumbar lordosis; PI, pelvic incidence; PT, pelvic tilt; SVA, sagittal vertical axis.

**Table 3 jcm-11-01020-t003:** Clinical parameters of the hybrid PF and open PF groups.

	Hybrid PF	Open PF	*p*-Value
VAS score for back pain			
Before surgery	57.3 ± 35.4	62.6 ± 29.0	0.576
3 months after surgery	29.2 ± 28.0	26.9 ± 30.7	0.598
Final follow-up	28.8 ± 24.7	25.0 ± 27.1	0.717
VAS score for leg pain			
Before surgery	50.8 ± 36.6	66.3 ± 30.0	0.144
3 months after surgery	25.3 ± 31.2	33.2 ± 32.7	0.456
Final follow-up	30.0 ± 31.2	29.5 ± 24.0	0.963
ODI			
Before surgery	55.4 ± 17.3	57.9 ± 16.6	0.631
3 months after surgery	44.3 ± 17.1	43.4 ± 15.0	0.867
Final follow-up	31.6 ± 20.3	30.0 ± 13.8	0.777

Values are presented as mean ± standard deviation, PF, posterior fixation; VAS, visual analog scale; ODI, Oswestry Disability Index.

## Data Availability

The data presented in this study are available on request from the corresponding author. The data are not publicly available due to privacy or ethical restrictions.

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
