# Peer review of "Comparison of Hybrid Posterior Fixation and Conventional Open Posterior Fixation Combined with Multilevel Lateral Lumbar Interbody Fusion for Adult Spinal Deformity"

_jcm, 2022, doi:10.3390/jcm11041020_

Round 1

Reviewer 1 Report

The manuscript “Comparison of hybrid posterior fixation and conventional open posterior fixation combined with multilevel lateral lumbar interbody fusion for adult spinal deformity” by Hirooki Endo, Hideki Murakami, Daisuke Yamabe, Yusuke Chiba, Ryosuke Oikawa, Hirotaka Yan and Minoru Doita aimed to evaluate the radiological and clinical results in ASD patients who underwent multilevel LLIF + hybrid PF compared to those that underwent multilevel LLIF + conventional open PF.

Below are my comments and remarks regarding the article:

1. The conclusions are inconsistent with the purpose of the study
2. Did the clinical results of the VAS ODI correlate with the sagittal balance?

Reviewer 2 Report

Thank you for giving me the opportunity to review the manuscript “Comparison of hybrid posterior fixation and conventional open posterior fixation combined with multilevel lateral lumbar interbody fusion for adult spinal deformity”(jcm-1556146).

The authors compared the multilevel LLIF combined with either hybrid posterior fixation or conventional open posterior fixation. They showed that with the minimal invasive hybrid technique, the estimated blood loss was significantly lower than during the conventional method. In addition, the MIS showed similar results regarding patients’ outcomes and radiological parameters.

Nevertheless, this study has some significant weaknesses:

  • First, the study’s design is retrospective and monocentric. Additionally, the study is biased by the different inclusion periods. During the first period, the patients were operated on with the conventional method, and during the second period, the remaining patients were operated on with the new hybrid MIS method.

  • The study’s novelty is limited. The new modifications used by the author represent an advancement of already established techniques. In addition, the significantly reduced estimated blood loss when using MIS has previously been shown by multiple studies.

  • The authors did not look for a shorter postoperative recovery using their new technique.

Additionally, there are some minor weaknesses:

  • There are some “hard to read” sentences (e.g., page 2 l. 63f).
  • Regarding the definition of PT, it is the center of both femur heads. So the plural “heads” should be used (page 3 l. 126)
  • The heading of Table 1 is incomplete as it only refers to the demographic data but not to the operative data.
  • Some references are missing and need to be discussed:
    1. Lai O, Chen Y, Chen Q, Hu Y, Ma W. Cadaveric biomechanical analysis of multilevel lateral lumbar interbody fusion with and without supplemental instrumentation. BMC Musculoskelet Disord. 2021 Mar 15;22(1):280. doi: 10.1186/s12891-021-04151-6. PMID: 33722233; PMCID: PMC7962251.
    2. Katz AD, Singh H, Greenwood M, Cote M, Moss IL. Clinical and Radiographic Evaluation of Multilevel Lateral Lumbar Interbody Fusion in Adult Degenerative Scoliosis. Clin Spine Surg. 2019 Oct;32(8):E386-E396. doi: 10.1097/BSD.0000000000000812. PMID: 30864972.
  • It would help understand that this study is of interest to the JCM readership if some JCM articles are cited. It should be possible to insert Matsukara et al. 2021 and Lombardi et al. 2021

Round 2

Reviewer 2 Report

I worked through the revisions and noticed that a lot of work had been done. As a result, the quality of the manuscript has been improved. The following comments are meant to help to improve the manuscript further.

Unfortunately, it was tough to follow the changes due to messed-up citations. The changes are linked to incorrect pages and lines that do not fit the merged pdf of the current version of the manuscript.

Regarding Response 1:

To my understanding, it is incorrect that multilevel LLIF with hybrid PF is not performed anywhere else. Katz et al. published a comparable study in 2019. Unfortunately, the authors failed again to highlight the novelty of their study.  

Additionally, as the same surgeons were responsible for the procedures from 2014 to 2018, it has not been discussed that the “lessons learned” from the open PF have led to a decreased complication rate for the hybrid PF.

I would like to see these details been added to M&M, especially because of the putatively biased learning curve as above-mentioned.

Regarding Response 2 + 3:

The additional inclusion of the CRP value is interesting. But patients do not necessarily benefit from a lower CRP value. For example, did the authors observe a faster postoperative mobilization of the hybrid subgroup than the open subgroup? Are there differences in the length of the hospital stay?

Regarding response 4 + 5:

All comments have been sufficiently addressed.  

Regarding response 6:

The recommendation was to change the heading of Table 1, not the content of the table. Demographic data only refers to age, income, race, education, etc., but not surgical data.

Response 7:

All comments have been sufficiently addressed.  
